# Urological, Digestive and Motor Function in Children After Prenatal or Postnatal Repair of Myelomeningocele

**DOI:** 10.3390/pediatric17060111

**Published:** 2025-10-22

**Authors:** Marianna Łoskot, Tomasz Koszutski

**Affiliations:** Department of Pediatric Surgery and Urology, Faculty of Medical Sciences in Katowice, Medical University in Silesia, 40-752 Katowice, Poland

**Keywords:** spina bifida, prenatal surgery, postnatal surgery, shunt-dependent hydrocephalus, musculoskeletal function, function of the urinary and digestive systems

## Abstract

**Objectives**: Myelomeningocele is one of the most common dysraphic defects. Does shortening the time of exposure to the toxic effects of amniotic fluid and mechanical trauma in utero on the herniated spinal cord and spinal nerves, thanks to prenatal surgery, reduce the risk of hydrocephalus with improved musculoskeletal function and better function of the urinary and digestive systems? The aim of the study was to compare the clinical effects of prenatal and postnatal surgery for myelomeningocele in pediatric patients. **Methods**: Comparison of urological, digestive and motor function in children following prenatal versus postnatal repair of myelomeningocele. The study group consisted of 110 children- 46 operated prenatally and 64 patients postnatally. Information about the children’s assessment of shunt-dependent hydrocephalus, motor skills, bladder and bowel function was obtained from a validated survey questionnaire completed by the children’s parents. **Results**: In the prenatal group, there was a significantly (*p* = 0.011) lower percentage of hydrocephalus treated with a shunt (45.71%) compared to the postnatal group (78.26%). The prenatal group revealed a lower percentage of paresis (*p* = 0.0422) and contractures of the lower limbs (*p* = 0.0108) and varus deformity (*p* = 0.0272). Also, in the prenatal group, fewer children move with only the use of a wheelchair (*p* = 0.0280) and more move independently or with orthopedic equipment (*p* = 0.0280). In prenatal children, the overall grade of vesico-ureteral reflux was significantly lower (*p* = 0.0105) and there was also a higher percentage of children with self-controlled defecation (*p* = 0.0395) and fewer children using enemas (*p* = 0.0269) and oral pharmacological agents (*p* = 0.0026). **Conclusions**: In children with myelomeningocele operated prenatally, compared to the postnatal group, shunt-dependent hydrocephalus and bladder and bowel incontinence were observed with significantly less frequency, and there was better musculoskeletal function. More children operated prenatally moved independently or with orthopedic equipment, and fewer used an orthopedic wheelchair. Further studies, particularly in even larger patient groups, are required to assess clinical benefits of prenatal surgery for children with myelomeningocele.

## 1. Introduction

Myelomeningocele (MMC) is the most common congenital neural tube defect resulting from incomplete closure of the posterior neuropore during primary neurulation, and develops in the early fetal life (days 21–28) [1,2]. MMC is the second most common disease after congenital heart defects and its incidence in the world is constantly increasing and amounts to 0.3–0.4 per 1000 live births, depending on sex, latitude and season of conception [3,4]. In Poland, the incidence of spina bifida is 0.29 cases per 1000, according to data from the Polish Registry of Congenital Malformations: Report 2021 [5]. The etiology of MMC is not clear, but it has been confirmed that folic acid deficiency or its metabolism in the mother are directly related to the occurrence of this defect in the fetus [3,6]. Risk factors for the occurrence of dysraphic defect in the child are diabetes and obesity of the mother, smoking, excessive alcohol consumption and the use of anticonvulsants (carbamazepine and valproic acid), antihistamine and sulfonamides drugs [4,7].

MMC is formed in the first weeks of fetal life during abnormal spinal cord roll-out and in the second stage when there is secondary damage to the spinal cord, protruding due to the toxic effects of amniotic fluid and mechanical injuries of the cord against the uterine wall (Heffez’s two-hit theory) [8]. Incomplete fusion of the spinal cord makes it vulnerable to chemical and mechanical stimuli, gradually causing its irreversible histopathological and functional damage. This leads, in most cases, to shunt-dependent hydrocephalus, impaired psychomotor development, lower extremity paresis and urinary and bowel incontinence [9]. The intensity of pathological symptoms in a child depends on the location of the damage (high-level MMC lesions) and the type of defect: MMC aperta or occulta [10], Figure 1. The use of fetal surgery (open or fetoscopic) allows for early protection of the nerve tissue from harmful factors and reduces the risk of disability with relatively low risk for the mother and fetus (the risk of premature birth) [11,12,13,14,15,16]. The aim of the study was to compare the clinical effects of prenatal and postnatal surgery for myelomeningocele in pediatric patients.

## 2. Materials and Methods

Study data were obtained by retrospective analysis of the clinical status of children with myelomeningocele (MMC), operated prenatally (*n* = 46) in the Department of Gynecology, Obstetrics and Oncological Gynecology in Bytom (Silesia), and 64 children operated after birth in the Department of Pediatric Surgery and Urology (of the Saint John Paul II Upper Silesian Child Health Centre in Katowice). Figure 2. Information about the children’s assessment of shunt-dependent hydrocephalus, motor skills, bladder and bowel function was obtained from a validated survey questionnaire (30 questions), completed by the children’s parents/guardians.

Participation in the survey was voluntary and anonymous, with the understanding that the information obtained would be used only for research purposes. The study was approved by the Bioethics Committee of the Medical University of Silesia in Katowice, Poland (No. PCN/CBN/0052/KB/185/22, date of approval: 4 October 2022).

### 2.1. The Study Subjects

The study group consisted of 110 children with MMC (aged 3 to 18 years): 46 children (median age—4.5 years) repaired in utero (30 girls and 16 boys) and 64 children (median age 6.0 years) operated postnatally (32 girls and 32 boys). In most patients (81.8%), myelomeningocele (MMC) occurred at the lumbo-sacral level of the spine. There was no significant correlation between gender (*p* = 0.1108), median age (*p* = 0.052) and the division of the current age of children under and over 7 years of age, Table 1, depending on the method of MMC closure, which indicates the proper selection of children with the comparability of both study groups. In the postnatal group, newborns were most often operated on in the first 24–48 h. In the prenatal group, the age at surgery was between 21 and 27 weeks of gestation, according to Johnson’s criteria [11]. Major inclusion criteria include single pregnancy, gestational age between 21 and 26 weeks, normal karyotype, no congenital anomalies, upper level of MMC damage below Th1 or above with the presence of Chiari II malformation in prenatal MRI, lateral ventricular size of MMC less than 17 mm, normal feto-placental function and voluntary consent to participate in the study. Major exclusion criteria include maternal age <18 years, uterine contractions, placental abnormalities, maternal infections (especially intrauterine) and other diseases (diabetes mellitus type 1, hypertension, obesity with a BMI > 28, thrombophilia).

### 2.2. The Survey Questionnaire

In order to conduct the research, a diagnostic survey method was used, for which a survey questionnaire was developed. The research tool was a validated survey questionnaire, consisting of a personal data sheet and 30 questions, mainly closed, (open-ended questions functioned as supplementary questions), addressed to parents/guardians of operated children. The main part of the questionnaire was divided into five parts: general information about the child’s health, the child’s neurological and orthopedic condition, the child’s mobility in the environment, the condition and functioning of the intestines and bladder, and performing daily activities.

The survey questionnaire was sent via social media to a selected group of parents of children operated on for myelomeningocele with the conscious use of “purposeful selection”, based on respondents’ general knowledge of the phenomenon under study. The study used a simplified survey validation method, which uses a “special filtering cafeteria” consisting of comparing the results obtained when the same respondents answered similar survey questions twice, which allows for the detection of inconsistencies in the answers or the assessment of the reliability of the answers.

Our survey contained mainly closed questions, so-called categorized (cafeteria) questions, allowing for the selection of answers from prepared proposals of a set of variants, the so-called cafeteria. This is the most commonly used type of question in quantitative research, where open-ended questions function as supplementary questions. Our survey is specific and original, so the assessment of its credibility (accuracy, consistency, reliability), i.e., validation, is also specific [17].

Natural randomness of selection was achieved by eliminating surveys that did not meet the adopted conditions of formal and substantive control. Formal (quantitative) control consisted of checking whether all forms had been completed, whether all items in the form had been included, whether all questions had been answered [17].

### 2.3. Statistical Analysis

The data were exported from an Excel v.2019 datasheet to the Statistica v. 13.1 data analysis system: StatSoft, Poland and the MedCalc v.23.0.9 package from MedCalc Software Ltd. (Ostend, Belgium) was used. Due to the lack of normal distribution of the investigated parameters (examined with the Shapiro–Wilk test), we used non-parametric tests such as the Mann–Whitney U test. In addition, we performed a chi-squared test (Chi—square.(X^2^) test NW) for the comparison of two small proportions (from independent samples), expressed as a percentage with significance level evaluation according to MLE-theory (Maximum Likelihood Estimation), and test for two structure indicators Model 2. The results are presented in the corresponding tables. All tables include the median for each sample, as well as the *p*-value for a given test. The level for statistical significance was set at a *p*-value < 0.05 [18,19].

## 3. Results

In the study, the duration of pregnancy was significantly shorter (*p* < 0.001) in mothers of children operated prenatally (median Hbd.—35.0) compared to the duration of pregnancy of mothers of children from the postnatal group (median Hbd.—38.0). Also, the birth weight of newborns operated prenatally (median grams—2035.0) was significantly lower (*p* < 0.0001) than in the postnatal group (median grams—3040.0). However, there were no significant differences (*p* = 0.0622) in the current weigh of children from the prenatal (median kilograms—16.5) and postnatal groups (median kilograms—19.3). The age of mothers of children undergoing prenatal surgery (median age—37.0 years) was similar (*p* = 0.077) to that of mothers’ age (median age—35.5 years) from the postnatal group, while fathers of children from the prenatal group (median age—41.0 years) were significantly older (*p* = 0.013) than fathers of children undergoing postnatal surgery (median age—37.0 years), Table 2. In the studied group of 110 children with myelomeningocele (MMC), a significant correlation with *p* = 0.0237 was found in the frequency of shunt-dependent hydrocephalus when comparing the group of children operated on prenatally (*n* = 46) with the postnatal group (*n* = 64), with a significantly (*p* = 0.011) lower percentage of hydrocephalus treated with a shunt in the prenatal group (45.71%) compared to the postnatal group (78.26%), Table 3. During the treatment of hydrocephalus, the shunt was replaced 17 times in the prenatal group and as many as 51 times in the postnatal group. Analysis of musculoskeletal function and motor skills in children with MMC showed a significantly lower percentage of paresis (*p* = 0.0422) and contractures of the lower limbs (*p* = 0.0108) and varus deformity (*p* = 0.0272) in children operated prenatally compared to the postnatal group. Similarly, in the prenatal group, a significantly higher percentage of operated children (*p* = 0.0292) did not have deformities of the lower limbs in the ankle joints. However, the percentage of spine deformities was not a factor differentiating both groups of children (*p* = 0.5693), Table 3. Despite a similar ability to move around independently in the environment (*p* = 0.2527) in both study groups, a significantly lower percentage of children moving only with the use of a wheelchair (*p* = 0.0286) and a higher percentage of children moving independently or with the use of orthopedic equipment (*p* = 0.0279) was found in the prenatal group compared to the postnatal group, Table 3. When assessing the bladder dysfunction in the two studied groups of children, no differences were found in the matter of self-controlled urination (*p* = 0.2902), urination by catheterization (*p* = 0.6269), urination with diapers (*p* = 0.1913) and urinary tract infections (*p* = 0.4607). Whereas in children operated prenatally, compared to the postnatal group, the overall grade of vesico-ureteral reflux was significantly lower (*p* = 0.0105), and importantly, the high-grade (IV–V) vesico-ureteral reflux was also significantly lower (*p* = 0.0454), Table 4. Comparison of bowel dysfunction in the prenatal group compared to the postnatal group showed a higher percentage of children with self-controlled defecation (*p* = 0.0395) and a lower percentage of children using enemas (*p* = 0.0269) and oral pharmacological agents (*p* = 0.0026), Table 4.

## 4. Discussion

The study included a total of 110 children with meningomyelocele (MMC), 46 of whom underwent in utero repair at the Department of Gynecology and Obstetrics in Bytom (Silesia), and 64 were operated on after birth at the Department of Pediatric Surgery and Urology in Katowice (Silesia).

In most of our patients (81.8%), myelomeningocele (MMC) occurred at the lumbosacral level of the spine. Similarly, Cope et al. showed that MMC most often occurs at the level of the lumbar spine (62%), and more rarely in the sacral (20%) and thoracic (18%) sections [20].

Up until 1997, the only option for MMC therapy was surgical correction after birth, and prenatal surgery procedures were possible only at the Children’s Hospital in Philadelphia (CHOP), based on Heffez’s theory (1995) [21,22]. The results of randomized studies (The Management of Myelomeningocele Study—MOMS) conducted in three American centers (Nashville, Philadelphia, San Francisco) demonstrated the clinical benefits of prenatal MMC surgery. These findings were subsequently corroborated by studies from European centers—Poland (Bytom, 2005), Switzerland (2010), Belgium and the Netherlands (2014)—which confirmed that fetal MMC surgery reduces the incidence of hydrocephalus and improves motor functions of the lower limbs [12,13,23,24,25].

In our study, children with MMC undergoing prenatal procedures showed a significantly (*p* = 0.011) lower percentage of hydrocephalus treated with a shunt (45.71%) compared to children from the postnatal group (78.26%). In the MOMS study, prenatal closure of a myelomeningocele reduced the risk of shunt-dependent hydrocephalus twice, from 82% in children operated after birth to 40% of children operated in the prenatal period [12]. An even smaller percentage of children with shunt-dependent hydrocephalus operated prenatally (28%) compared to the postnatal group (80%) was found in the Polish report from 2014 [13]. The most recent diagnostic MRI study also demonstrated differences in neuro-developmental anatomy, justifying reduction of hydrocephalus and improvement of neurological symptoms in MMC patients treated prenatally versus postnatally [26]. Partially different results based on the anatomical, radiological and clinical features of patients with MMC showed that children after hydrocephalus treated prenatally were more likely to develop hydrocephalus after 3 months of life than those after postnatal repair, although the overall rate of hydrocephalus was higher in the postnatal group [27]. Reducing the incidence of shunt-dependent hydrocephalus is associated with a reduced risk of obstruction of the intraventricular or peripheral shunt, mechanical failure of the shunt itself, and complications from the peritoneal cavity [12,13].

In our study group during the treatment of hydrocephalus, the shunt was replaced 17 times in the prenatal group and three times more often—51 times—in the postnatal group. Also, the Polish results (2020) of neonatal examinations after prenatal repair indicate good efficacy in reducing hindbrain herniation and reducing the prevention of cerebral ventricle enlargement [16].

In children after MMC surgery with shunt-dependent hydrocephalus, additionally, segmental disorders in the innervation of the intestines and urinary bladder occur, causing chronic constipation and urinary incontinence due to the inability to maintain proper pressure in the bladder and its regular emptying (bladder and bowel syndrome) [14,26]. In previous publications, it has been shown that children with MMC operated prenatally, compared to the postnatal group, have both better social urinary continence with fewer urinary tract infections and fewer complaints due to chronic constipation [14,28]. Better maintenance of social continence for urine (social dryness) in children after fetal repair is the effect of better-preserved functions of the urogenital diaphragm muscles, especially the external urethral sphincter muscle [29]. Additionally, fewer constipations mean better complete emptying of the urinary bladder with less chronic urine retention and therefore a lower risk of urinary tract infections [28,30]. When assessing the bladder dysfunction in our studied groups of children, no significant differences were found in the matter of self-controlled urination, urination by catheterization (CIC) and diapers as well as in the incidence of urinary tract infections. An important positive difference between both studied groups is the significantly less common overall grade of vesico-ureteral reflux, and especially a lower incidence of high-grade vesico-ureteral reflux (IV-V) in children operated on prenatally. In the latest publication comparing slightly younger children (approximately 2 years of age) operated prenatally and postnatally due to MMC, similarly, no statistically significant differences were found between both groups when analyzing clean intermittent catheterization (CIC), the use of anticholinergic drugs and self-controlled urination [31].

Our children from the prenatal group showed better self-control of bowel movements and used enemas and oral pharmacological agents less frequently than the postnatal group. The lower incidence of constipation in children operated in utero may probably be the result of having a longer section of properly innervated large intestine due to the earlier separation of the spinal cord and spinal nerves from toxic amniotic fluid and mechanical injuries in utero. Additionally, the enormous plastic and regenerative capabilities of the cells of the nervous system in fetuses affect the maintenance and formation of proper neuronal pathways, with improved functioning of both the digestive and urinary systems and lower limb motor skills [30,32].

Shunt malfunctions and urinary tract infections were the most common reasons for unanticipated hospitalizations in children and youth with MMC [33]. Moreover, hydrocephalus, renal failure and scoliosis are common causes of mortality in patients after MMC surgery [34].

Our study of musculoskeletal function and motor skills revealed that significantly fewer children in the prenatal group had paresis and contractures of the lower limbs and varus deformity, as well as fewer deformities of the lower limbs in the ankle joints. However, the percentage of spine deformities was not a factor differentiating both groups of children. Despite a similar ability to move around independently in the environment in both study groups, a significantly lower percentage of children from the prenatal group move only with the use of a wheelchair and, more importantly, more of these children move independently or with the use of orthopedic equipment. The ability to move independently, participation in recreation, with better urinary and bowel function is considered by most children with MMC after correction and their parents as one of the most important elements improving their quality of life [15,35,36]. In order to improve the quality of life of children with MMC, in addition to a higher level of mobility and emotional and social function, good family support and a high financial status are important [37,38,39].

Treatment of MMC after correction is multidisciplinary and involves at the same time radiologists, neurologists, orthopedists, urologists, gastroenterologists, psychologists and bioethical experts [40,41,42]. However, it should always be remembered that in every therapeutic choice, the best interest of the child must be paramount.

European organizations have established consensus and guidelines (SHINE Conference-Belfast 2017) for patients with MMC [43]. The new 2020 guidelines address the care of patients with spina bifida, particularly with regard to rehabilitation therapy [44,45,46].

Prenatal repair MMC is an invasive procedure; however, the risk of maternal–fetal surgery and the risk of premature birth is not large compared to the enormous potential to improve the quality of life of children [16,43,47]. The professional experience of the surgical team performing prenatal MMC surgery is very important in reducing the risk of complications.

In our study, children undergoing prenatal surgery had a shorter gestation period and lower birth weights compared to the postnatal group. However, the differences in current body weight were not statistically significant.

Our research showed a better therapeutic effect of prenatal surgery compared to postnatal surgery in children with MMC. In the prenatal group, there was less shunt-dependent hydrocephalus and better musculoskeletal function: paralysis and bilateral contractures, varus deformity of the lower limbs and deformation of lower limbs in the ankle joints. Fewer children undergoing prenatal surgery used an orthopedic wheelchair because they were able to move independently or with orthopedic equipment. Bladder and bowel dysfunction were also less common in the prenatal group: vesico-ureteral reflux, especially high grade, greater self-controlled defecation, defecation with enemas and oral pharmacological agents.

Our positive therapeutic effects of prenatal surgery are consistent with the results of other authors [14,15,29,30], but these reports usually narrowly compared the functioning of individual systems, most often the neurogenic bladder or musculoskeletal system. In addition, recent publications have focused on the functioning of children with MMC only, after prenatal or fetoscopy procedures, without any postnatal surgery.

The novelty of our study is the holistic and comprehensive comparison of the functioning of the urological, digestive and musculoskeletal systems, both in children after prenatal or postnatal surgery for myelomeningocele. These are novel and original findings in children with MMC and no similar data have been found in available publications. We are very grateful to all parents of children with MMC who, despite the daily challenges of caring for their disabled children, found the time and strength to complete our survey.

A limitation of our publication is the small sample size—in the case of a very rare defect, it is difficult to obtain a large group of respondents, which limits the reliability and generalizability of the results. Initially, the number of surveys obtained from mothers of children from the prenatal group was 57, of which, after conducting formal and substantive control, only 46 surveys were qualified for statistical evaluation, while the number of questionnaires obtained from mothers of children operated postnatally was 78, of which, after the control, only 64 surveys were qualified.

Therefore, in our conclusions we emphasized the need for further research, preferably multi-center, to increase the reliability and generalizability of our results.

The increasing knowledge of the positive effects of prenatal surgeries among medical professionals and parents of children with myelomeningocele raises hopes for an increase in the number of European and global medical centers with the possibility of performing prenatal surgeries.

## 5. Conclusions

In children with myelomeningocele operated prenatally, compared to the postnatal group, shunt-dependent hydrocephalus and bladder and bowel incontinence were observed significantly less frequently, and there was better muscosceletal function. More children operated prenatally moved independently or with orthopedic equipment and fewer patients used an orthopedic wheelchair. Further studies, particularly in even larger patient groups, are required to assess clinical benefits of prenatal surgery for children with myelomeningocele.

## Figures and Tables

**Figure 1 pediatrrep-17-00111-f001:**
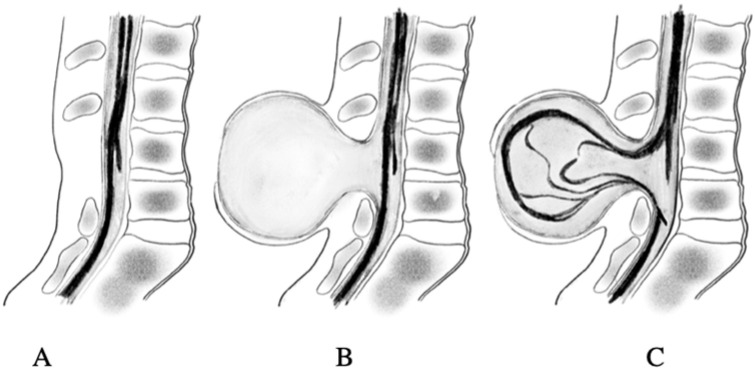
Types of defect oculta: (**A**) spina bifida oculta, (**B**) meningocele, (**C**) myelomeningocele. Authors’ source.

**Figure 2 pediatrrep-17-00111-f002:**
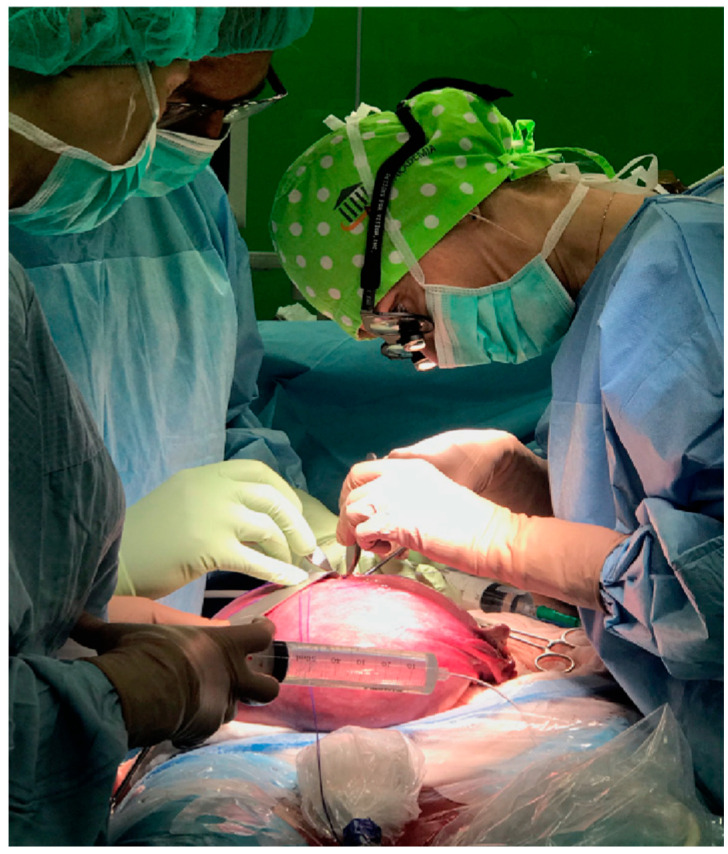
Prenatal open-uterine MMC surgery by a team of pediatric surgeons from Katowice (Silesia), Poland. Authors’ source.

**Table 1 pediatrrep-17-00111-t001:** Comparison of current age of children under and over 7 years of age with MMC operated prenatally (Prenatal Group) and postnatally (Postnatal Group).

Method of Surgery Myelomeningocele	Current Age of Children		
	<7 Years	≥7 Years	Total
Prenatal Group	35	11	46
% Column	49.30%	28.21%	
% Row	76.09%	23.91%	
Postnatal Group	36	28	64
% Column	50.70%	71.79%	
% Row	56.25%	43.75%	
Total	71	39	110
Chi^2^, Yates, test: *p* = 0.0520			

**Table 2 pediatrrep-17-00111-t002:** Comparison of the clinical characteristics in children with myelomeningocele operated prenatally (Prenatal Group) and postnatally (Postnatal Group).

	Total Cohort (*n* = 110)	Prenatal Group (*n* = 46)	Postnatal Group *(n* = 64)	Test	*p*-Value
Age (median)	5.5	4.5	6.0	Chi^2^, Yates	0.0520
Female, *n (%)*	62 (56%)	30 (48%)	32 (52%)	Chi^2^, NW	0.1110
Male, *n (%)*	48 (44%)	16 (33%)	32 (67%)	Chi^2^, NW	0.1108
Mother’s age, (median)	37.0	37.0	35.5	Mann–Whitney U test	0.0773
Father’s age, (median)	39.0	41.0	37.0	Mann–Whitney U test	0.0133
Duration of pregnancy, (Hbd; median)	37.0	35.0	38.0	Mann–Whitney U test	<0.001
Birth weight, (grams; median)	2735.0	2035.0	3040.0	Mann–Whitney U test	0.0001
Current body weight, (kilograms, median)	18.0	16.5	19.3	Mann–Whitney U test	0.0622
Myelomeningocele at the lumbosacral level of the spine	90 (81.82%)	39 (84.78%)	51 (79.69%)	Chi^2^, NW	0.7520

**Table 3 pediatrrep-17-00111-t003:** Comparison of the musculoskeletal function and motor skills in children with myelomeningocele operated prenatally (Prenatal Group) and postnatally (Postnatal Group).

	Total Cohort (*n* = 110)	Prenatal Group (*n* = 46)	Postnatal Group *(n* = 64)	Test	*p*-Value
Shunt-dependent hydrocephalus	52(47.3%)	16 (45.71%)	36 (78.26%)	Chi^2^, NW Test for two structure indicators Model 2	0.0024 0.0011
Paralysis of the lower limbs	79 (71.82%)	29 (63.04%)	50 (78.13%)	Test for two structure indicators Model 2	0.0422
Bilateral contractures of the lower limbs	71 (64.55%)	24(52.17%)	47(73.44%)	Test for two structure indicators Model 2	0.0108
No deformation of lower limbs in the ankle joints	39 (35.45%)	21 (45.65%)	18 (28.13%)	Test for two structure indicators Model 2	0.0292
Varus deformity of lower limbs	50 (45.45%)	16 (34.78%)	34 (53.13%)	Test for two structure indicators Model 2	0.0272
Deformities of the spine	56 (50.91%)	22 (47.83%)	34 (53.13%)	Chi^2^, test Yatesa	0.5693
Moving in wheelchair only	32 (29.09%)	9 (19.57%)	23 (35.94%)	Test for two structure indicators Model 2	0.0286
Moving independently or with the use of orthopedic equipment (hybrid)	78 (70.91%)	37 (80.43%)	41 (64.06%)	Test for two structure indicators Model 2	0.0279
Moving independently	38 (34.55%)	17 (36.96%)	21 (32.81)	Chi^2^, NW	0.2527

**Table 4 pediatrrep-17-00111-t004:** Comparison of the Bowel and Bladder Dysfunction (BBD) in children with myelomeningocele operated prenatally (Prenatal Group) and postnatally (Postnatal Group).

	Total Cohort (*n* = 110)	Prenatal Group (*n* = 46)	Postnatal Group (*n* = 64)	Test	*p*-Value
Urinating with diapers	63 (57.27%)	23 (50.0%)	40 (62.50%)	Test for two structure indicators Model 2	0.0956
Urination by catheterization (CICI)	91 (82.73%)	39 (84.78%)	52 (81.25%)	Test for two structure indicators Model 2	0.3130
Self-controlled urination	5 (4.54%)	1 (2.17%)	4 (6.25%)	Test for two structure indicators Model 2	0.1393
Vesico-ureteral reflux (VUR) all grade	23 (20.91%)	5 (10.87%)	18 (28.13%)	Test for two structure indicators Model 2	0.0105
Vesico-ureteral reflux (VUR)—high grade (IV–V)	9 (8.18%)	1 (2.17%)	8 (12.50%)	Chi^2^, test NW	0.0454
Urinary tract infections	58 (52.72%)	24 (52.17%)	34 (53.13%)	Test for two structure indicators Model 2	0.4607
Self-controlled defecation	66 (60.0%)	32 (69.57%)	34 (53.13%)	Test for two structure indicators Model 2	0.0395
Constipations—defecation with enemas	19 (17.27%)	9 (19.57%)	10 (15.63%)	Chi^2^, test NW	0.0269
Constipations—defecation with oral pharmacological agents	21 (19.09%)	4 (12.50%)	17 (40.48%)	Test for two structure indicators Model 2	0.0026

## Data Availability

The data presented in this study are available on request from the corresponding author due to legal reason -Regulation (EU) 2016/679 of the European Parliament and of the Council of 27 April 2016 on the protection of natural persons with regard to the processing of personal data and on the free movement of such data, and repealing Directive 95/46/EC (General Data Protection Regulation).

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
