# Peer review of "Urological, Digestive and Motor Function in Children After Prenatal or Postnatal Repair of Myelomeningocele"

_pediatrrep, 2025, doi:10.3390/pediatric17060111_

Round 1
Reviewer 1 Report
Comments and Suggestions for Authors
There are several instances where the first letter of a word is missing- "hunt" instead of "shunt", "espondents" instead of "respondents," both in the abstract and paper. These all must be corrected, please.
The wording of the first sentence should be reconsidered in order to be scientifically accurate- spina bifida and myelomeningocele are, strictly, speaking, not synonymous- myelomeningocele is a type of spina bifida.
Without being able to see the referenced tables , it is a little difficult to assess the effectiveness of the conclusions completely, but in general using solely parent survey data without being able to validate that data with more standardized objective will be weighed differently by different readers of the paper.
The median age was stated of patients surveyed- what was range of followup? Would be helpful to know in survey type of paper. ( It might be in the tables but would be nice to have in narrative.)
The discussion section could be better divided into paragraphs for ease of reading.
In the discussion there are references to " in numerous publications...". as well as very concrete statements of fact topics that are still being studied, particularly how fetal surgery affects bladder and bowel function, so one might want to keep that in mind when crafting and wording sentences.
Very nice sentence about expressing gratitude to families who completed survey! Would also add comment about limitations of this kind of data, however.
Author Response
Manuscript ID pediatrrep-3820419
Reviewer # 1
Dear Reviewer,
We thank you very much for the new insightful review and valuable suggestions.
Below, we explain point by point the new revised details of the corrections in the manuscript:
1. “There are several instances where the first letter of a word is missing- "hunt" instead of
"shunt", "espondents" instead of "respondents," both in the abstract and paper. These all
must be corrected, please”
.
Our answer:
We sincerely apologize for any spelling errors found in this work. All errors have been carefully
identified and corrected. Thank you for bringing them to our attention.
2. “The wording of the first sentence should be reconsidered in order to be scientifically
accurate- spina bifida and myelomeningocele are, strictly, speaking, not synonymous-
myelomeningocele is a type of spina bifida.”
Our answer:
Of course, you are right, meningoencephalocele is the most advanced form of spina bifida, which we
illustrated in Figure 1. We used a mental shortcut unnecessarily, but there is already a correction in
red in the first words of chapter 1 "Introduction" as follows:
“Myelomeningocele (MMC) is the most common congenital neural tube defect resulting from
incomplete closure of the posterior neuropore during primary neurulation, develops in the early fetal
life (days 21-28) [1, 2]”.
3. Without being able to see the referenced tables, it is a little difficult to assess the
effectiveness of the conclusions completely
Our answer:
The initial version of the publication included three tables that we referenced in the text, but which,
presumably due to a transmission error, did not reach the Editorial Office. We sincerely apologize for
this. We are including the tables (Table 2, Table 3, Table 4) in Chapter 3 "Results" in their improved
form in the revised publication.
4. The median age was stated of patients surveyed- what was range of follow up?
Would be helpful to know in survey type of paper. ( It might be in the tables but would be nice
to have in narrative.)
Our answers:
As you wisely clinically suggested, we have also added Table 1 in Materials and Methods chapter
(subchapter "The Study Subjects) and short texts in red as follows:
”The study group consisted of 110 children with MMC (aged 3 to 18 years): 46 children (median age –
4.5 years) repaired in utero (30 girls and 16 boys) and 64 children (median age 6.0 years) operated
postnatally (32 girls and 32 boys). In most patients (81,8%) myelomeningocele (MMC) occurred at the
lumbosacral level of the spine. There was no significant correlation between gender (p=0.1108),
median age (p=0.052) and the division of the current age of children under and over 7 years of age
Table 1, depending on the method of MMC closure, which indicates the proper selection of children
with the comparability of both study groups”.
All tables are attached to the review.
5. but in general using solely parent survey data without being able to validate that data with
more standardized objective will be weighed differently by different readers of the paper.2
Manuscript ID pediatrrep-3820419
Our answers:
The entire statistical analysis of our work was performed by a very competent, experienced
practitioner and theoretician in the field of medical sciences, a professor and lecturer at the Silesian
University, who helped us justify the choice of this simple validation of our survey with a double-
checking version of the answers by the same respondent in a homogeneous group of mothers of
children operated on for meningoencephalocele.
At the end of subchapter The Survey Questinaire, we included information about our survey
validation method as follows:
”The study used a simplified survey validation method, which uses a "special filtering cafeteria"
consisting in comparing the results obtained when the same respondents answered similar survey
questions twice, which allows for the detection of inconsistencies in the answers or the assessment of
the reliability of the answers .
Our survey contained mainly closed questions, so-called categorized (cafeteria) questions, allowing
for the selection of answers from prepared proposals of a set of variants, the so-called cafeteria. This
is the most commonly used type of question in quantitative research, where open-ended questions
function as supplementary questions. Our survey is specific and original, so the assessment of its
credibility (accuracy, consistency, consistency, reliability), i.e. validation, is also specific [17].
Natural randomness of selection was achieved by eliminating surveys that did not meet the adopted
conditions of formal and substantive control. Formal (quantitative) control consisted of checking
whether all forms had been completed, whether all items in the form had been included, whether all
questions had been answered” [17].
As we prepare our next survey publication, we'll conduct additional statistical consultations, based
on your suggestion.
6. In the discussion there are references to " in numerous publications...". as well as very
concrete statements of fact topics that are still being studied, particularly how fetal surgery
affects bladder and bowel function, so one might want to keep that in mind when crafting
and wording sentences.
Our answers:
I wanted to thank you very much for teaching me "scientific humility." I'm a younger generation of
doctors, having just graduated from medical school a few years ago. I was so pleased with the
consistency of my survey results with the reports of other authors that I over-interpreted the
ongoing research results. Following your suggestion, I changed the text of the sentence in the
Discussion chapter:
"In previous publications it has been shown that children with MMC operated prenatally, compared to
the postnatal group, have both better social urinary continence with fewer urinary tract infections
and fewer complaints due to chronic constipation [14, 28].
"
In my defense, I can cite the last sentence from the Conclusions section.
7. The discussion section could be better divided into paragraphs for ease of reading.
Our answers:
As you suggested, we improved the text of the Discussion chapter, dividing it into thematic
paragraphs to make it easier for readers.
Thank you very much for this comment.3
Manuscript ID pediatrrep-3820419
8. Very nice sentence about expressing gratitude to families who completed survey! Would also
add comment about limitations of this kind of data, however.
Our answer:
Thank you for your suggestion. According to it, we added some sentences to the Discussion chapter
about limitations as follows:
“A limitation of our publication is the small sample size – in the case of a very rare defect, it is
difficult to obtain a large group of respondents, which limits the reliability and generalizability of the
results. Initially, the number of surveys obtained from mothers of children from the prenatal group
was 57, of which, after conducting formal and substantive control, only 46 surveys were qualified for
statistical evaluation, while the number of questionnaires obtained from mothers of children
operated postnatally was 78, after the control, only 64 surveys were qualified.
Therefore, in our conclusions, we emphasized the need for further research, preferably multi-
center, to increase the reliability and generalizability of our results.”
Dear Reviewer,
We have tried very hard to respond to your suggestions, and we sincerely hope that you will
appreciate our efforts and be satisfied with our responses.
We really appreciate it. .Thanks for these suggested changes; our article has gained new value.
Best regards,
Authors

Reviewer 2 Report
Comments and Suggestions for Authors
The article addresses a topic of great clinical and scientific interest, adopting a holistic approach that is particularly valuable because it considers urological, digestive, and motor aspects. The authors aim to compare the clinical outcomes of children operated on for myelomeningocele prenatally with those operated on postnatally. The retrospective research design is sound, with a well-structured comparison between the two groups and adequate statistical analysis.
However, the manuscript requires substantial revision in form and structure. First, there are frequent formatting errors, font changes, and an inconsistent layout, making the text less readable.
The bibliographic citations do not follow a uniform style or correct format, and, for a topic of international relevance, the number of sources is limited. It would be appropriate to expand the literature review to include recent studies and international guidelines.
A section dedicated to the limitations of the study is missing; it would be desirable for the authors' considerations to be included in the article.
It would also be helpful if the authors included tables summarizing the results and diagrams that clearly illustrate the research design.
I would like to emphasize that the study is scientifically sound and potentially highly relevant (particularly noteworthy is the discussion of the results, which extends the research findings with the international literature and explores potential future clinical developments, once again emphasizing the integrated and systemic approach between medical specialties and various healthcare disciplines), but it requires careful editorial review and critical analysis to fully meet the standards required of the journal.
Author Response
1
Manuscript ID pediatrrep-3820419
Reviewer # 2
Dear Reviewer,
We thank you very much for the new insightful review and valuable suggestions.
First, we wanted to express our gratitude and joy for so many warm words of appreciation. Never
before has a reviewer lavished so much praise on our publications. We sincerely thank you.
This is a tremendous encouragement and motivation for us to continue our scientific work.
And now, .below, we explain point by point the new revised details of the corrections in the
manuscript:
1. However, the manuscript requires substantial revision in form and structure. First, there are
frequent formatting errors, font changes, and an inconsistent layout, making the text less
readable. The bibliographic citations do not follow a uniform style or correct format.
Our answer:
We sincerely apologize for frequent formatting errors, inconsistent layout and fond changes in our
work. All errors in form and structure have been carefully identified and corrected.
We also took great care to correct errors in the format and standarization of bibliographic
references. Thank you for bringing them to our attention.
2. for a topic of international relevance, the number of sources is limited. It would be
appropriate to expand the literature review to include recent studies and international
guidelines.
Our answer:
Of course, you are right, the consensus and European guidelines are published in 2017, as you
suggested, we have added three newer (2020, 2021) references (no. 44, 45, 46) in the Discussion
chapter:
44. Sawin KJ, Brei TJ, Houtrow AJ. Quality of life: Guidelines for the care of people
with spina bifida. J Pediatr Rehabil Med. 2020;13(4):551-564.
45. .Blount JP, Bowman R, Dias MS, et al. Neurosurgery guidelines for the care of
people with spina bifida. J Pediatr Rehabil Med. 2020;13(4):467–477.
46. Starowicz J, Cassidy C, Brunton L. Health Concerns of Adolescents and Adults
with Spina bifida. Front Neurol.2021;12:667665.
with the following sentence:
“European organizations have established consensus and guidelines (SHINE Conference-Belfast 2017)
for patients with MMC [43]. The new 2020 guidelines address the care of patients with spina bifida,
particularly with regard to rehabilitation therapy [44-46].”
as well as two new reports from 2024 (no. 38 and 39):
38. Kaluri AL, Jiang K, Abu-Bonsrah N, Ammar A, et al. Socioeconomic characteristics
and postoperative outcomes of patients undergoing prenatal vs. postnatal repair
of myelomeningoceles. Child’s Nervous System. 2024;40(4):1177-1184.
39. Talamonti G. Reflections upon the intrauterine repair of myelomeningocele.
Child’
s Nerv Syst. 2024;40(6):1571–1575.
3. A section dedicated to the limitations of the study is missing; it would be desirable for the
authors' considerations to be included in the article.2
Manuscript ID pediatrrep-3820419
Our answer:
Thank you for your thoughtful comment. According to it, we added some sentences to the Discussion
chapter about limitations as follows:
“A limitation of our publication is the small sample size – in the case of a very rare defect, it is difficult
to obtain a large group of respondents, which limits the reliability and generalizability of the results.
Initially, the number of surveys obtained from mothers of children from the prenatal group was 57, of
which, after conducting formal and substantive control, only 46 surveys were qualified for statistical
evaluation, while the number of questionnaires obtained from mothers of children operated
postnatally was 78, after the control, only 64 surveys were qualified.
Therefore, in our conclusions, we emphasized the need for further research, preferably multi-center,
to increase the reliability and generalizability of our results.”
4. It would also be helpful if the authors included tables summarizing the results and
diagrams that clearly illustrate the research design.
Our answer:
Of course, You are right. The initial version of our publication included three tables that we
referenced in the text, but which, presumably due to a transmission error, did not reach the Editorial
Office. We sincerely apologize for this. We are including the tables (Table 2, Table 3, Table 4) in
Chapter 3 "Results" in their improved form in the revised publication, along with one additional table
(Table 1) in Materials and Methods chapter.
All tables are attached to the review.
Dear Reviewer,
Once again, thank you very much for the new review and we hope that our new answers are also
satisfactory to you.
We really appreciate it. Thanks to these changes, our article gained new value.
Best regards,
Authors

Round 2
Reviewer 2 Report
Comments and Suggestions for Authors
Thank you very much for the revisions and the detailed responses to all the comments. The changes regarding formatting, references, and the addition of new literature and limitations section are greatly appreciated.
However, the tables you mention (Tables 1–4) still do not appear to be included in the revised manuscript. Could you kindly check and ensure that they are properly attached or integrated into the text?
Thank you again for your efforts in improving the manuscript.
Author Response
Dear Reviewer,
Thank you very much for your valuable comments on my manuscript – I truly appreciate them.
With regard to Tables 1–4, I would like to clarify that they were submitted as a separate file, in accordance with the journal’s submission guidelines. I understood that their integration into the text is the responsibility of the editorial office. If I am mistaken, I apologize for the oversight.
To avoid any inconvenience, I am resubmitting Tables 1–4 as an attachment to this response to your review.
Sincerely,
Authors
